# HyperRAG: Query-Centric Retrieval-Augmented Generation with Hyperbolic Structuring

## Abstract

Retrieval-Augmented Generation (RAG) has demonstrated significant potential in enhancing large language models (LLMs) by supplementing external knowledge. However, existing approaches focus primarily on retrieving isolated factual knowledge entities while neglecting the critical reasoning relationships. To address this limitation, Graph Retrieval-Augmented Generation (GraphRAG) has emerged as an effective solution, which explicitly integrates structured knowledge graphs into LLMs to support complex reasoning tasks. Although diverse corpus retrieval methods have been explored, they typically rely on static, query-agnostic graphs constructed via fixed heuristics. We are thereby motivated to propose a query-centric retrieval framework that adaptively constructs a graph tailored to each query. However, it is challenging to accurately identify these latent relationships from queries to the corpus. Moreover, unifying multiple local-perspective connections into a globally coherent structured corpus introduces additional complexity. To this end, we introduce HyperRAG, a novel framework in the Hyperbolic space that captures both explicit entity-based links and implicit logical connections inferred by the LLM. Our main contributions include: $(i)$ A dual-stage prompting strategy that guides the LLM to identify relevant passages and their implicit relationships based on the query. $(ii)$ A hierarchical graph unification paradigm that models each query-specific graph as a minimal subtree and integrates them into a cohesive graph. $(iii)$ A hyperbolic space embedding approach that effectively preserves the hierarchical structure during graph learning. Extensive experiments have been conducted on three benchmark datasets, where a remarkable improvement on three datasets indicates our superior performance compared with others.

## 1 Introduction

Recent advances in retrieval-augmented generation (RAG) have demonstrated its importance by integrating external knowledge to improve the downstream performance of large language models (LLMs) Gao et al. (2024); Lewis et al. (2020a). While effective for fact-based queries, conventional RAG systems only retrieve fragmented passages without capturing the rich semantic relations Dong et al. (2023); Fatehkia et al. (2024); Liang et al. (2024). This limitation becomes particularly apparent in domains requiring multi-hop reasoning or conceptual understanding, where the connections between knowledge are as crucial as the ideas themselves.

The emergence of graph retrieval-augmented generation (GraphRAG) represents an effective solution, augmenting retrieval with structured knowledge representations as a graph Zhang et al. (2025); Xiao et al. (2025). Current approaches to graph construction in GraphRAG systems can be broadly categorized into three paradigms. Passage graphs offer computational efficiency through entity-based linking but sacrifice semantic depth Li et al. (2024). Hierarchical trees provide multi-scale organization yet remain rigid in their predefined structures Sarthi et al. (2024). Knowledge graphs excel at factual reasoning but are constrained by their schema-dependent nature Edge et al. (2024); He et al. (2024); Luo et al. (2025). Yet, these methods are limited by their static and query-independent graph construction, often relying on pre-defined heuristics such as entity co-occurrence or clustering. These methods fail to adapt to the unique reasoning paths required by individual queries and overlook the opportunity to extract task-relevant, latent relations embedded in natural language. As shown in Figure 1, a typical graph built on a Disney-related corpus often groups entities based on general categorical features (e.g., grouping animated films by the topic). These structures fail to answer

query-specific questions such as *"Which film was released right after the Disney Renaissance period by Disney?"* since they do not capture latent, query-aware relations.

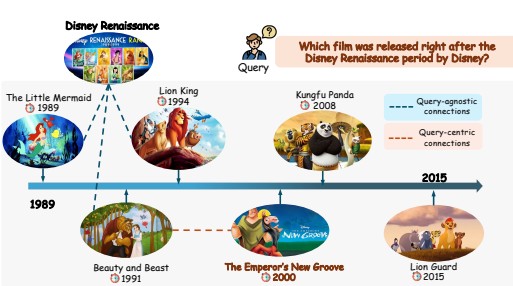

Figure 1: An illustration of the difference between query-agnostic and query-centric graph construction. In the former one, passages are connected due to shared themes or categories. But the second one connects passages guided by the question itself. This example well demonstrates our motivation to achieve a coherent fusion of two types of relationships from both perspectives.

Therefore, we are motivated to model the latent relations among concepts by adequately exploiting the query-specific relations for a more knowledgeable GraphRAG. However, this remains challenging for two major reasons. First, it is difficult to translate from abstract query semantics to concrete knowledge in large corpora. It requires overcoming the inherent ambiguity and sparsity of natural language. Second, it is hard to integrate all the query-specific knowledge into the unified graph. This paper addresses the research question:

***In what ways can query-specific implicit knowledge be leveraged to enhance GraphRAG?***

To this end, we introduce HyperRAG, a novel query-centric framework that adapts to the reasoning requirements of each question. For the query above, the graph is built starting with relevant background, and then selectively incorporates passages through meaningful linking. $(i)$ We propose a dual-stage prompting strategy that first identifies explicit entity-level connections and then extracts query-specific implicit relations using LLM-based reasoning. This enables our method to integrate both general background knowledge and fine-grained logical cues. $(ii)$ Each query-specific graph is naturally modeled as a minimalistic subtree. A hierarchical graph unification paradigm is proposed where all the subtrees are cohesively integrated into a unified one. $(iii)$ To maximally preserve the hierarchical properties, the graph learning is conducted in the hyperbolic space, given its superior performance in modeling tree structures.

**Contributions are summarized as follows:**

- We propose a novel query-centric graph construction paradigm, which contains both static knowledge relationships and underlying logical connections guided by the query itself. This allows the constructed graph to adapt flexibly to different reasoning needs.
- We employ a dual-prompting approach to select useful passages and incorporate local connections into a unified graph. Combined with our hyperbolic graph learning framework, it better captures hierarchical and semantic structures than current methods.
- Extensive experiments have been conducted on three benchmark datasets, where a remarkable improvement indicates our superior effectiveness and further confirm the suitability of adopting hyperbolic distance for representing hierarchical and logically structured relationships.

## 2 PRELIMINARY

**Notations.** Let $\mathcal{Q} = q_1, q_2, \ldots, q_N$ denote the set of input questions, and let $\mathcal{C} = p_1, p_2, \ldots, p_M$ represent the corpus containing $M$ textual passages. For each question $q_i \in \mathcal{Q}$, we construct a query-specific graph $G_i = (V_i, E_i, S_i)$, where $V_i \subseteq \mathcal{C}$ is the set of relevant passages (nodes), $E_i \subseteq V_i \times V_i$ is the set of undirected edges connecting semantically related passages, and $S_i : V_i \to \mathbb{T}$ maps each node to its corresponding textual content. A text encoder $f : \mathbb{T} \to \mathbb{R}^d$ (e.g., Sentence-Bert) maps both passages and questions into a $d$-dimensional embedding space. These embeddings are further projected into a Poincaré ball hyperbolic space $\mathbb{H}^d$, where retrieval is performed based on the hyperbolic distance between the query embedding and the nodes in $V_i$.

Retrieval-Augmented Generation (RAG) has become a popular paradigm for enhancing language models with external knowledge, especially in open-domain question answering and long-form generation tasks. Current RAG methods often treat knowledge bases as flat collections, retrieving top-K passages based on embedding similarity. While often effective, this approach struggles with complex, domain-specific, or multi-hop queries. Recently, GraphRAG has gained increasing popularity. Instead of storing knowledge as isolated passages, it organizes them into a graph

structure, where nodes represent passages and edges encode various forms of semantic or contextual relationships. This shift allows the retrieval process to go beyond simple similarity links in the graph. A key insight behind this approach is that the organization of knowledge significantly influences what can be retrieved. In GraphRAG, the graph serves not only as memory but also as a guide for relationships. As a result, the construction of the graph is crucial.

# 3  HyperRAG: Query-centric Retrieval-augmented Generation

Our primary goal is to construct a query-guided graph over a textual corpus to enhance retrieval-augmented generation performance. Instead of relying on a predefined knowledge graph, our method dynamically builds a graph that is tailored to each input question. This allows the structure of the graph to reflect the dependency relationships most relevant to the query. Existing approaches based on knowledge graphs often suffer from limited coverage and irrelevant connections, where relationships are predefined and typically represented in the form of triples. They often result in information loss by compressing rich textual content into sparse symbolic facts. In contrast, our approach preserves the full semantics of the original corpus passages and leverages LLMs to reason about which passages are useful for a given question, which ones should be linked, and what additional information might be needed. This query-specific graph, constructed in hyperbolic space, serves as a flexible and expressive structure for guiding hierarchical and semantically-aware retrieval, ultimately improving the quality of generated answers in downstream tasks.

## 3.1  Query-centric Graph Construction from Raw Text

### 3.1.1  Building Explicit Knowledge Connection

To complement query-specific reasoning, we incorporate an explicit knowledge connection module that emulates the relational structure found in traditional knowledge graphs. In knowledge graphs, entities such as "*Arthur Conan Doyle*", "*Sherlock Holmes*", and "*Dr. Watson*" are linked through well-defined relations like writes, assistant of, and so on. While symbolic triples (head, relation, tail) enable structured reasoning, they often rely on external rules, as illustrated in the format below:

$$(p_i, r, p_j) \in \mathcal{T}, \quad \text{where } \mathcal{T} = \{(h, r, t) \mid h, t \in \mathcal{V}, r \in \mathcal{R}\}.$$

Inspired by this, we define explicit connections by identifying shared or semantically similar entities across different passages. Using lightweight keyword extraction techniques, we detect key phrases such as named entities, concepts, or domain-specific terms and connect passages that reference the same or closely related terms. This results in a set of basic, query-independent links that reflect common knowledge relationships within the corpus. As shown below, $\mathcal{K}(p)$ denotes the keyword set of passage $p$, and $\theta$ is a threshold set empirically to $0.15$. To prevent the graph from being dominated by dense explicit edges, we restrict each passage to be connected to at most 3 neighbors through co-occurrence links. This constraint ensures the explicit structure remains at a reasonable scale.

$$K_i = \{k_1^{(i)}, k_2^{(i)}, \ldots, k_m^{(i)}\}, \quad K_j = \{k_1^{(j)}, k_2^{(j)}, \ldots, k_n^{(j)}\}.$$

$$\text{sim}(p_i, p_j) = \frac{|K_i \cap K_j|}{\min(|K_i|, |K_j|)}, \quad e_{ij} \in \mathcal{E}_{\text{exp}} \quad \text{if} \quad \text{sim}(p_i, p_j) \geq \theta.$$

These explicit edges form a foundational layer of the graph, providing entity-level connectivity similar to that of traditional knowledge graphs. Unlike hard-coded triples, however, our connections retain the full textual context and allow for flexibility in expression. Later in the framework, these explicit links are integrated with the query-guided implicit connections to form a richer, hybrid graph structure that captures both general background knowledge and question-specific reasoning paths.

### 3.1.2  Building Query-guided Implicit Connection

While explicit connections provide basic entity-level linkage across the corpus, they remain independent of any specific information need. To introduce query relevance into the graph structure, we propose a query-guided implicit connection mechanism. The key idea is to leverage the reasoning capabilities of LLMs to infer logical relationships between passages in the context of a specific

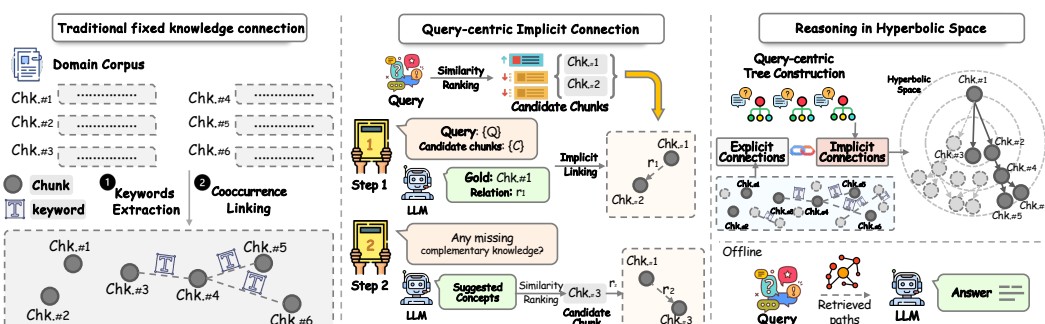

Figure 2: The overall framework of our proposed HyperRAG. We begin by preprocessing the corpus using keyword extraction techniques to identify and connect similar passages, often grouped by shared topics or entities. Then, for each question, we find out the underlying logical connections among passages. These connections are embedded into a hyperbolic space to naturally reflect the hierarchical tree-structured nature. Finally, we retrieve the top-K items to support question answering.

question. Given a query $q$, we first retrieve a candidate set of passages $\mathcal{P} = \{p_1, p_2, \ldots, p_K\}$ from a large corpus $\mathcal{C}$ using a dense retrieval Sentence-Bert-Based model. Each passage and the query are embedded into a semantic space, and we select the top-$K$ passages.

We then prompt the LLM to evaluate which of these passages are potentially helpful for answering the query, and return a subset of passages deemed relevant. Based on this filtered set, the LLM is further asked to reason about their interrelations, i.e., whether two passages support, complement, or expand upon each other in answering $q_i$. The identified relationships are used to construct implicit edges between passages, forming a query-specific subgraph that reflects not only content similarity, but also task-driven semantic alignment. To enrich this reasoning process, we also ask the LLM to suggest missing concepts or complementary knowledge that may aid in answering the question. These suggestions are converted into additional queries to retrieve further passages from the corpus, which are then incorporated into the evolving graph. As a result, the implicit connection module dynamically builds a graph structure that is tailored to the query and driven by LLM-based understanding rather than predefined rules or rigid entity matching.

### 3.1.3 FUSION OF BOTH EXPLICIT AND IMPLICIT EDGES

To construct a unified hierarchical structure that combines both factual and query-specific relationships, we propose a tree fusion mechanism that merges the **explicit** and **implicit** knowledge trees into a single tree structure. The explicit tree is derived from surface-level cues, such as co-occurring entities or shared keywords among corpus chunks, capturing predefined and globally relevant knowledge relations. In contrast, the implicit tree is constructed dynamically for each query $q$ by leveraging a language model to identify semantically useful passages and connect them based on their contextual relevance to $q$. Given two sub-trees that exhibit explicit and implicit dependency relationships between textual segments, we perform fusion by first identifying shared nodes across both trees, i.e., nodes that reference the same passage or contain the same key entity. These shared nodes are unified into a single node in the resulting graph $\mathcal{G}$, and their respective subtrees are recursively merged while preserving the original parent-child directionality. To ensure that the resulting structure remains a valid tree, we prevent the formation of any cycles by discarding redundant edges that introduce loops. The final tree thus integrates both global knowledge priors and query-dependent reasoning chains, offering a rich and hierarchically coherent structure suitable for downstream retrieval tasks.

### 3.2 RETRIEVAL WITH LEARNED HYPERBOLIC EMBEDDING FROM CONSTRUCTED GRAPH

To faithfully encode the hierarchical and query-sensitive structure of our fused graph $\mathcal{G} = (\mathcal{V}, \mathcal{E})$, a key design choice is how to define the distance between nodes that capture both semantic and structural information. To this end, we introduce a hyperbolic embedding strategy specifically tailored for Retrieval-Augmented Generation over structured corpora. We embed all nodes (i.e., passages) into the Poincaré ball $\mathbb{B}^d = \{x \in \mathbb{R}^d : \|x\| < 1\}$, where distances increase exponentially with radius, allowing for separation of semantically close versus distant nodes Nickel & Kiela (2017).

---

**Algorithm 1:** Streaming Query-Centric Graph Construction and Retrieval

---

**Input:** Query $q$, corpus $\mathcal{C}$, retrieval size $K$, max rounds $T$, graph $\mathcal{G} = (\mathcal{V}, \mathcal{E})$
**Output:** Updated graph $\mathcal{G}'$, answer $a$

1 **for** $t \leftarrow 1$ **to** $T$ **do**
2    **if** *NeedMoreContext*$(\mathcal{R}_{t-1}, q)$ **then**
3      $\mathcal{P}_t \leftarrow$ Retrieve$(\mathcal{R}_{t-1}, \mathcal{G}, K)$;
4      $\mathcal{R}_t \leftarrow$ SelectRelevant$(\mathcal{P}_t, q)$;
5      $\mathcal{V} \leftarrow \mathcal{V} \cup \mathcal{R}_t$;
6      $\mathcal{E} \leftarrow \mathcal{E} \cup$ BuildEdges$(\mathcal{R}_t, \mathcal{R}_{t-1})$;
7    **end**
8 **end**
9 **if** *GetQueryCount*$() \geq$ *update_frequency* **then**
10    $\mathcal{V}_{\text{new}} \leftarrow$ GetRecentNodes$(\mathcal{G})$;
11    $\mathcal{G}' \leftarrow$ UpdateEmbeddings$(\mathcal{G}, \mathcal{V}_{\text{new}})$;
12    $\mathcal{G}' \leftarrow$ PruneOldNodes$(\mathcal{G}')$;
13    ResetQueryCount$()$;
14 **end**
15 $a \leftarrow$ GenerateAnswer$(q, \mathcal{G})$;
16 **return** $\mathcal{G}', a$

---

**Motivation for Hyperbolic Distance.** Common metrics include $L^p$ norms, each with different characteristics Coghetto (2016); Debeye & Van Riel (1990). While desirable in many metric spaces, these metrics universally satisfy the triangle inequality. As the number of hops increases, passages with similar semantic embeddings have endpoints $p_0$ and $p_k$ that remain close in the metric space. As $k$ increases, the global semantic distance between distant nodes in the chain is underestimated. This becomes a problem in reasoning tasks where longer chains should reflect logical difference or inferential effort. For instance, passages separated by multiple reasoning steps shouldn't be treated as nearly the same as directly connected ones. Please refer to Appendix A.1 for detailed discussion and mathematical illustrations. Therefore, we choose hyperbolic geometry for two reasons:

- **Exponential Expansion:** Euclidean space's linear growth cannot accommodate all nodes in deep trees. As a result, many logically irrelevant passages become spatially indistinguishable, because the space becomes "crowded". On the contrary, hyperbolic space is particularly suitable for modeling hierarchical tree structures due to its exponentially expanding space. This inherent capacity ensures that hierarchical relationships are encoded without losing spatial coherence. Let $\mathbb{R}^n$ and $\mathbb{H}^n$ denote $n$-dimensional Euclidean and hyperbolic spaces, respectively. The volume of a ball of radius $r$ differs significantly between the two geometries, and it is mathematically shown as follows:

$$D = \sum_{i=0}^{k-1} \delta(p_i, p_{i+1}), \ \text{Vol}_{\mathbb{R}^n}(r) = \frac{\pi^{n/2}}{\Gamma(n/2+1)} r^n, \ \text{Vol}_{\mathbb{H}^n}(r) = \int_0^r \sinh^{n-1}(t)dt \sim C \cdot e^{(n-1)r}.$$

- **Logically Aligned Proximity:** In multi-hop QA, we often reason over chains of passages $\{p_0, p_1, \ldots, p_k\}$. Nodes in a parent-child logical relation (e.g., $p_i \rightarrow p_{i+1}$) can remain at small distances, while nodes far apart in reasoning hierarchy (e.g., $p_0 \rightarrow p_k$) can be naturally separated.

$$d_{\mathbb{H}}(p_0, p_1) \approx d_{\mathbb{H}}(p_1, p_2) \approx \cdots \approx d_{\mathbb{H}}(p_{k-1}, p_k) \ll d_{\mathbb{H}}(p_0, p_k).$$

Thus, logical edges are modeled with short hyperbolic distances, preserving the immediate inferential relationship between supporting facts. In contrast, nodes that are farther apart in the reasoning hierarchy, which may only share indirect or topic-level connections, are naturally pushed apart due to the exponential expansion of hyperbolic space. This creates a geometric alignment between reasoning depth and embedding distance, which is essential when organizing logic-driven passages.

**Joint Learning of Semantic and Structural Representations.** We initialize the Bert-based embeddings and optimize them such that connected nodes in the graph are mapped closer in hyperbolic space. The loss function minimizes the sum of pairwise Poincaré distances between adjacent nodes: $\mathcal{L} = \sum_{(u,v) \in \mathcal{E}} d_{\mathbb{B}}(x_u, x_v)$, where $x_u, x_v \in \mathbb{B}^d$ are the hyperbolic embeddings of nodes $u$ and $v$, and $d_{\mathbb{B}}(\cdot, \cdot)$ denotes the Poincaré distance. Intuitively, this distance measures how far apart two points are in a curved hyperbolic space. It can be viewed as a projection of a negatively curved space onto the interior of a Euclidean ball. As points move closer to the boundary, they become exponentially farther away, which is defined as:

$$d_{\mathbb{B}}(x, y) = \text{arcosh}\left(1 + \frac{2\|x - y\|^2}{(1 - \|x\|^2)(1 - \|y\|^2)}\right).$$

This loss encourages embeddings of connected passages to move closer together while naturally preserving the global hierarchical structure due to the curvature of the space. Unlike Euclidean

distance, the hyperbolic metric enforces stronger discrimination at the boundary, allowing the model to preserve fine-grained semantic relationships even with limited dimensions. This property is particularly beneficial when passages are connected via both implicit and explicit edges, enabling more meaningful retrieval of leaf passages.

**Streaming Hyperbolic Representation Learning & Retrieval.** To adapt to the evolving information needs and continuously improve retrieval performance, we implement a streaming update mechanism for our hyperbolic embeddings. As new queries are processed over time, they are efficiently incorporated into the existing hyperbolic space without requiring a full retraining from scratch. This incremental learning approach leverages the inherent capacity of hyperbolic geometry to accommodate new hierarchical structures and semantic relationships. By periodically updating the embeddings based on recent query patterns, our system dynamically refines its representation of the knowledge corpus, ensuring that the retrieval remains both accurate and relevant to the latest interactions. This streaming paradigm enables our model to maintain its performance edge in real-world, evolving environments. Based on this, we leverage the learned geometry to retrieve relevant information for question answering. Specifically, given a query $q$, we first encode it into a hyperbolic embedding $\mathbf{z}_q \in \mathbb{B}^d$ using the same sentence encoder followed by projection into the Poincaré ball. The goal is to identify leaf nodes in the graph, i.e., raw passages from the corpus that are closest to the query in hyperbolic distance. At inference time, we compute the hyperbolic distance between $\mathbf{z}_q$ and all leaf node embeddings $\{\mathbf{z}_i\}$, where each $\mathbf{z}_i \in \mathbb{B}^d$ represents a passage. According to the experimental trials, we typically select three candidate passages for retrieval and then feed these along with the question into the language model in the form of a prompt to generate the answer. By retrieving in hyperbolic space, we benefit from the hierarchical structure of the graph, which allows for efficient identification of semantically and structurally relevant content, even across multi-hop relationships.

## 4 EXPERIMENTS

We conduct experiments on three public open datasets: HotpotQA Yang et al. (2018), 2WikiMultiHopQA Ho et al. (2020), and the Musique Trivedi et al. (2022b) dataset. Utilizing the content retrieved by hyperRAG, we employ the GPT-4o-mini language model to answer a variety of questions. This enables us to assess the effectiveness of our framework in leveraging retrieved information to enhance the quality and relevance of responses generated by the language model. Our study aims to address the following questions: (1) In what ways does HyperRAG compare to existing state-of-the-art answer generation pipelines in terms of performance? (2) How does our method demonstrate its time and token-level efficiency compared to others? (3) How do different settings influence the performance of our framework? (4) How can we intuitively distinguish between hyperbolic and Euclidean distances, and how are these patterns reflected in our empirical data?

### 4.1 EXPERIMENTAL SETTINGS

**Datasets.** We utilize three datasets for our experiments, each comprising 1,000 questions sourced from their respective domains. Each dataset's corpus contains approximately ten thousand independent chunks of information, allowing for a diverse range of queries and contexts. These datasets collectively represent progressively complex reasoning scenarios, spanning diverse domains from basic scientific terminology to humanities subjects, providing a comprehensive evaluation of our method's capability to handle various complexity levels in knowledge-intensive tasks. To ensure the consistency and fairness of the experiments, we strictly maintain the same questions and corpus as utilized in previous studies Gutiérrez et al. (2024).

**Implementation Details.** Our experiments were conducted on a server equipped with four RTX-4090 GPUs. We employ both single-step and multi-round retrieval strategies based on the constructed graph, which facilitates fast and accurate retrieval performance. During model training within the hyperbolic space, we utilize the Adam optimizer with an initial learning rate of 0.01. The hyperparameters, such as the number of epochs and retrieval items, are tuned through a grid search, with the best values chosen for each dataset. We also record the time and token consumption.

**Baselines.** To comprehensively evaluate the effectiveness of our proposed framework, we compare it with a diverse set of baselines spanning several paradigms. These methods are categorized into three groups: the first includes direct language model approaches that use pre-trained models like GPT or Llama to answer questions through zero-shot prompting without additional retrieval mechanisms;

Table 1: Performance comparison among state-of-the-art baselines and `HyperRAG` on three benchmark datasets in terms of both String-Match and GPT-evaluation Accuracy.

| Model | HotpotQA | | 2Wiki | | Musique | |
|---|---|---|---|---|---|---|
| | Match-Acc. | GPT-Acc. | Match-Acc. | GPT-Acc. | Match-Acc. | GPT-Acc. |
| **Direct Zero-shot LM Inference** | | | | | | |
| Llama3 (8b) Touvron et al. (2023) | 10.8 | 11.6 | 12.4 | 9.2 | 3.9 | 4.8 |
| Llama3 (13b) | 9.6 | 7.7 | 13.1 | 10.6 | 4.7 | 4.4 |
| GPT-3.5 | 28.3 | 39.8 | 27.3 | 31.6 | 13.2 | 17.9 |
| GPT-4o-mini Achiam et al. (2023) | 30.4 | 32.1 | 31.0 | 33.9 | 12.5 | 18.3 |
| **Retrieval-augmented Variants** | | | | | | |
| IRCoT Trivedi et al. (2022a) | 45.5 | 48.9 | 35.4 | 38.7 | 19.1 | 22.4 |
| Retrieval (Top-1) | 38.4 | 42.6 | 34.8 | 37.3 | 13.2 | 18.5 |
| Retrieval (Top-3) | 43.2 | 45.1 | 41.2 | 43.5 | 16.6 | 19.2 |
| Retrieval (Top-5) | 44.1 | 45.9 | 40.7 | 42.4 | 16.9 | 19.8 |
| **Graph-enhanced Generation Methods** | | | | | | |
| KGP Wang et al. (2024) | 46.4 | 57.1 | 41.5 | 43.7 | 23.3 | 27.3 |
| G-retriever He et al. (2024) | 41.3 | 40.9 | 26.7 | 25.7 | 14.1 | 15.6 |
| LightRAG Guo et al. (2024) | 47.8 | 52.7 | 46.3 | 43.3 | 28.3 | 27.7 |
| HippoRAG (single-step) | 46.8 | 50.9 | 44.6 | 45.6 | 21.2 | 24.3 |
| HippoRAG (multi-step) Gutiérrez et al. (2024) | 53.7 | 55.6 | 49.7 | 49.2 | 32.0 | 31.8 |
| GFM-RAG Luo et al. (2025) | 55.1 | 56.2 | 48.6 | 50.8 | 29.3 | 32.6 |
| RAPTOR Sarthi et al. (2024) | 48.1 | 55.3 | 47.7 | 43.9 | 28.2 | 29.7 |
| `HyperRAG` (single-step) | 54.3 | 57.5 | 47.8 | 51.8 | 28.7 | 31.4 |
| `HyperRAG` (multi-step) | **57.4** | **60.9** | **50.8** | **53.3** | **34.7** | **36.4** |

the second group consists of enhanced LM methods that incorporate auxiliary techniques such as similarity-based retrieval and chain-of-thought prompting to improve reasoning capabilities; the third category involves graph-based augmented generation approaches that leverage structured knowledge graphs, including methods based on common knowledge graphs and hierarchical clustering strategies.

**Evaluation Metrics.** For evaluation metrics, existing methods primarily rely on string matching techniques to assess answer accuracy. The most commonly adopted metrics include Exact Match, which requires the predicted answer to exactly match the ground truth string, and Answer Containment, which checks whether the ground truth answer is contained within the model's prediction. However, the correctness of answers can sometimes be overlooked if the phrasing differs, even when the answer is correct. To address this, GPT-based evaluation methods have gradually been adopted. We utilize a dual evaluation strategy comprising both string-matching and LLM-based judgment to achieve a fair and clarifying comparison. This is motivated by the need to overcome the limitations of any single metric, balancing objective reproducibility with nuanced semantic assessment.

## 4.2 MAIN RESULTS

The overall comparison between HyperRAG and other baselines is presented in Table 1. As mentioned above, the baselines are categorized into three groups. From the zero-shot performance, it is evident that advanced LLMs already possess considerable answering capability, reflecting a strong storage of background knowledge. Additionally, similarity-based retrieval and chain of thought prompting significantly improve performance, demonstrating that incorporating supplementary information through multi-round interaction effectively enhances answer quality. Thirdly, overall, graph-based variants tend to outperform purely semantic similarity-based methods. This improvement stems from the graph structure's ability to enhance reasoning capacity by explicitly modeling relationships and dependencies. However, not all graph RAG approaches are equally effective. Sometimes, they do not surpass direct retrieval methods. This highlights the importance of how the graph is constructed: the quality and relevance of the graph structure are crucial factors that determine whether the graph-based approach will yield better results. In practice, we employ two distinct paradigms for answer generation: single-round and multi-round reasoning. For straightforward queries, a single retrieval and reasoning cycle suffices to produce accurate answers. For more complex questions requiring multi-hop reasoning, the model first decomposes them into simpler sub-questions, performs sequential retrieval for each sub-question, and then synthesizes the intermediate results to form a comprehensive final answer. Empirical results have shown that incorporating additional information retrieved in hyperbolic space significantly improves the model's performance, as it provides richer, more relevant context that helps the model better understand and address complex queries.

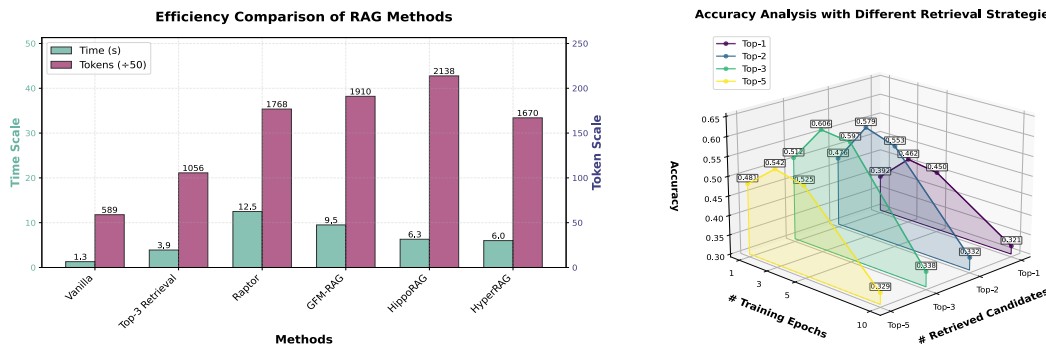

Figure 3: (Left) Time and token consumption; (Right) Effect of hyperparameters.

### 4.3 Analyzing the Computational Efficiency of HyperRAG

Our HyperRAG framework introduces two major sources of computational cost. The first occurs during query-aware graph construction, where we reason over the relationships between the input question and its retrieved candidate passages. For a dataset of 1,000 queries and around 10,000 passages, the full graph construction and multi-step answering take approximately 6 seconds per query. However, the inference phase demonstrates significantly higher efficiency when employing a single-round approach. Once the graph is constructed, the system only requires a lightweight search and reasoning step without iterative model interactions, making it suitable for practical real-life use. In this optimized setup, each query processing time is reduced to approximately 3 seconds, encompassing both retrieval and answer generation. Although the current implementation processes tasks iteratively, the framework can be further accelerated by integrating multi-processing techniques.

For a more comprehensive evaluation, we compare HyperRAG against a range of baselines, including zero-shot LLM, similarity-based dense retrieval, and several graph-based retrieval variants on the left of Figure 3. As expected, the zero-shot method is the fastest since it bypasses retrieval entirely. Dense retrieval approaches, such as BERT-based retrievers, offer a good tradeoff with slightly higher latency but much better answer quality. GraphRAG-style models exhibit the most variability in runtime due to differences in graph construction strategies, the number of retrieval hops, and the multi-step inference. HyperRAG incurs only a modest computational cost compared to simple retrieval while being efficient than other graph-based counterparts. By leveraging a fast hyperbolic retrieval technique and capped-round reasoning, our model achieves highly competitive efficiency.

### 4.4 Effects of Semantic & Structural Representation Learning

In our framework, the ultimate passage representations result from a combination of initial semantic embeddings and structural refinement through hyperbolic space learning. We analyze how the number of training epochs in hyperbolic representation learning affects performance on the right of Figure 3. This process essentially controls the tradeoff between semantic fidelity and structural alignment. When the number of training epochs is too small, the learned embeddings remain largely similar to the initial BERT-based representations, preserving strong semantic similarity but failing to encode deeper relational structure across the reasoning graph. In contrast, when trained for too many epochs, the representations become dominated by the graph's structural connections. Our experiments show that setting the number of hyperbolic training epochs to 3 provides a good balance. Another important factor affecting performance is the number of retrieved passages during inference. Interestingly, we observe that retrieving more candidates does not always help. When the number of retrieved passages exceeds 5, the overall performance tends to decrease. This suggests that adding too much irrelevant content may hinder the model's ability to focus on the most useful evidence. Besides, we test how important the explicit and implicit edges are by conducting an ablation study. Based on the results shown in Table 2, we observe that explicit edges contribute to a modest performance improvement across all datasets, while implicit edges significantly enhance the model's effectiveness. These results confirm that explicit and implicit connections play distinct roles in the reasoning process. Explicit edges effectively capture surface-level topical associations and structural relationships, providing a foundational context. Conversely, implicit edges are crucial for tracing the underlying logical pathways required to answer complex questions, enabling deeper and adaptive inference.

### 4.5 COMPARISON OF HYPERBOLIC AND EUCLIDEAN DISTANCES

To gain deeper insights into the properties of hyperbolic geometry in embedding spaces, we randomly sampled queries from the dataset and systematically compared their distance distributions to the corpus embeddings in both hyperbolic and Euclidean spaces. The phenomenon of larger hyperbolic distance values can be explained by the exponential expansion property of hyperbolic geometry. Under the same embedding dimensionality, hyperbolic space provides a more expansive representation space, enabling pushing unrelated entities to greater distances. This distance distribution characteristic offers two key advantages. **Enhanced Discriminative Power**: The larger distance numerical range provides richer gradient signals for the model, facilitating learning of more precise similarity decision boundaries. **Improved Ranking Stability**: The amplification effect of distance differences makes retrieval rankings more stable, reducing sorting uncertainties in borderline cases.

<table>
<tr><td colspan="4">Table 2: Effects of Connection Strategies</td></tr>
<tr><td>Method</td><td>HotpotQA</td><td>Wiki</td><td>Musique</td></tr>
<tr><td>Semantic retrieval</td><td>45.1</td><td>43.5</td><td>19.2</td></tr>
<tr><td>Explicit edges</td><td>48.8</td><td>44.9</td><td>21.3</td></tr>
<tr><td>Implicit edges</td><td>55.2</td><td>50.0</td><td>29.9</td></tr>
<tr><td>HyperRAG</td><td>57.5</td><td>51.8</td><td>31.4</td></tr>
</table>

Table 2: Effects of Connection Strategies

| Method | HotpotQA | Wiki | Musique |
|---|---|---|---|
| Semantic retrieval | 45.1 | 43.5 | 19.2 |
| Explicit edges | 48.8 | 44.9 | 21.3 |
| Implicit edges | 55.2 | 50.0 | 29.9 |
| HyperRAG | 57.5 | 51.8 | 31.4 |

Table 3: Comparison of Distance Metrics

| Metric | Hyperbolic Distance | | | Euclidean Distance | | |
|---|---|---|---|---|---|---|
| | Max | Min | Mean | Max | Min | Mean |
| Top-1 | 2.22 | 1.59 | 1.87 | 1.11 | 0.29 | 0.71 |
| Top-3 | 2.23 | 1.56 | 1.97 | 1.33 | 0.29 | 0.87 |
| Top-10 | 2.29 | 1.45 | 2.10 | 1.44 | 0.27 | 1.06 |

A critical challenge in high-dimensional Euclidean space is the distance concentration phenomenon François et al. (2007). The squared Euclidean distance asymptotically follows a normal distribution. This leads to poor discrimination between nearest and furthest neighbors. In contrast, the logarithmic growth of the arcosh function for large arguments and the amplification of small coordinate differences near the boundary induce a heavy-tailed distance distribution. This distribution is right-skewed, providing superior discriminative power for learning semantic hierarchies, as evidenced in Table 3.

## 5 RELATED WORK

Retrieval-augmented generation (RAG) has become a prominent paradigm for open-domain and multi-hop question answering, where an external corpus is indexed and queried to retrieve relevant documents that are then fed into a generative model Lewis et al. (2020b). Early approaches such as REALM Guu et al. (2020) and DPR Sachan et al. (2021) focus on encoding large text corpora into dense embeddings, enabling scalable and differentiable retrieval. Subsequent work improved retrieval-augmented generation by incorporating fusion mechanisms or editable memory Bajaj et al. (2022); Hofstätter et al. (2023). To better capture the semantic and logical relationships between passages, some studies explored mind map-style structures that model paragraph connectivity through co-reference, discourse, or logical links. Building on this intuition, GraphRAG approaches explicitly incorporate graph structures to enrich the retrieval process, offering additional reasoning paths or relational priors to assist generation Edge et al. (2024); Dong et al. (2025); Zhou et al. (2025). Among them, HippoRAG Gutiérrez et al. (2024) and GFM-RAG Luo et al. (2025) are examples of KG-based methods. HippoRAG leverages external knowledge graphs to enhance context understanding during retrieval and generation. GFM-RAG goes further by constructing and completing knowledge graphs from text, aiming to improve downstream generation through enhanced graph learning. In contrast, RaptorRAG Sarthi et al. (2024) avoids reliance on external knowledge bases by applying hierarchical clustering over the corpus. This yields a multi-level document structure, allowing queries to navigate the corpus in a coarse-to-fine manner. In summary, most existing methods construct static graphs either through predefined knowledge graphs or heuristic relations between passages.

## 6 CONCLUSION

We observe that knowledge passages are connected not only through explicit relationships but also through implicit logical relationships that emerge only under specific questions. To this end, we introduce a novel approach that fuses both perspectives into a single graph, enabling a query-centric retrieval. Moreover, by recognizing the inherent tree-like structure of multi-hop reasoning, we get rid of traditional Euclidean representations and instead learn embeddings in a hyperbolic space, which better preserves hierarchical relationships and logical distance. Compared with traditional KG-based and graph-based models, our method demonstrates superior performance. Such advantages highlight its potential for practical deployment in knowledge-intensive applications.

ETHICS STATEMENT

This research utilizes exclusively publicly available benchmark datasets (e.g., HotpotQA, Wiki, Musique) that contain no personally identifiable information or private user data. All datasets employed have been previously published for academic research purposes with appropriate ethical oversight. Our work poses no additional ethical risks beyond those inherent in the original dataset collections, as we do not collect, annotate, or process any new human-subject data. The experimental procedures involve standard natural language processing tasks that do not raise ethical concerns regarding privacy or potential harm to individuals.

REPRODUCIBILITY STATEMENT

To ensure the reproducibility of our findings, we have provided comprehensive details throughout this paper: (1) All baseline implementations are based on established methods with clear citations to original sources; (2) Experimental configurations including hyperparameters, training details, and evaluation metrics are thoroughly documented; (3) The core framework is presented in the main text with sufficient implementation details. The complete code and data processing scripts will be made publicly available upon publication.

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

APPENDIX

## THE USE OF LARGE LANGUAGE MODELS (LLMS)

In this work, LLMs (specifically GPT-4) were employed solely for limited textual refinement tasks, primarily for paraphrasing technical descriptions and improving the fluency of certain passages. All LLM-generated content was carefully reviewed and refined by the authors to ensure technical accuracy and alignment with our original scientific contributions. The core intellectual contributions, methodological innovations, and scientific insights remain entirely author-driven.

## A  MORE DETAILS ABOUT HYPERGRAPH

### A.1  DISTANCE CHOICE FOR GRAPH-BASED REASONING

After constructing a passage graph $\mathcal{G} = (\mathcal{V}, \mathcal{E})$, a key design choice is how to define the semantic distance between nodes. Common metrics include $L^p$ norms, each with different characteristics. Let $x, y \in \mathbb{R}^n$ be embeddings of two passages. The general $L^p$ norm is defined as:

$$d_p(x, y) = \left( \sum_{i=1}^{n} |x_i - y_i|^p \right)^{1/p}. \tag{1}$$

There are some popular variants of the general $L^p$ norm family, each offering a different geometric intuition and sensitivity to feature dimensions. The $L^1$ norm (Manhattan distance) emphasizes coordinate-wise differences, the $L^2$ norm (Euclidean distance) captures straight-line distance in space, and the $L^\infty$ norm (Chebyshev distance) reflects the maximum single-coordinate deviation. These distances are commonly used for computing similarity between item embeddings. Mathematical expressions are presented in the following:

$$\text{Manhattan Distance:} \quad d_1(x, y) = \sum_{i=1}^{n} |x_i - y_i| \quad .$$

$$\text{Euclidean Distance:} \quad d_1(x, y) d_2(x, y) = \left( \sum_{i=1}^{n} (x_i - y_i)^2 \right)^{1/2}. \tag{2}$$

$$\text{Chebyshev Distance:} \quad d_\infty(x, y) = \max_i |x_i - y_i|.$$

While desirable in many metric spaces, these metrics universally satisfy the triangle inequality. As the number of hops increases (i.e., $p_1 \to p_2 \to \cdots \to p_k$ ), passages with similar semantic embeddings have endpoints $p_0$ and $p_k$ that remain close in the metric space. As $k$ increases, the global semantic distance between distant nodes in the chain is underestimated.

$$d(x, z) \leq d(x, y) + d(y, z). \tag{3}$$

$$d(p_0, p_k) \leq \sum_{i=0}^{k-1} d(p_i, p_{i+1}). \tag{4}$$

This becomes a problem in reasoning tasks where longer chains should reflect increased difference or inferential effort. For instance, passages separated by multiple reasoning steps shouldn't be treated as nearly the same as directly connected ones. Euclidean-like metrics end up "flattening" the structure, losing sight of the underlying hierarchy or compositional relationships within the information.

Besides, hyperbolic space enables a large number of nodes per parent, reflecting the branching nature of trees. Additionally, distances in hyperbolic space show logarithmic growth with respect to similarity, quantified by

$$d(u, v) \approx - \log(\text{similarity}(u, v)). \tag{5}$$

Furthermore, hyperbolic spaces can effectively embed high-dimensional trees, where a tree with a branching factor $b$ and depth $h$ contains

$$N = \frac{b^{h+1} - 1}{b - 1} \quad \approx \quad \frac{b^{h+1}}{b - 1} \sim O(b^h) \tag{6}$$

nodes. The constant negative curvature of hyperbolic spaces facilitates the efficient packing of nodes, allowing for more nodes in a given volume while preserving hierarchical relationships. For instance, the Poincaré disk model visually represents these properties, illustrating how distances correspond to relationships in a tree structure. Together, these features underscore hyperbolic space's suitability for accurately capturing the complexity of hierarchical tree structures.

## A.2 CASE STUDY OF QUERY-CENTRIC GRAPH CONSTRUCTION

### A.2.1 IMPLICIT CONNECTION

As shown below, while passage #1 explicitly defines the temporal boundary of the Disney Renaissance period (1989–1999), it does not itself mention any subsequent films. However, passages #4, #5, and #6 provide information about films and series released after this era. The connection between passage #1 and these later passages is implicit, as they are linked not by surface-level semantic similarity but by a shared temporal and logical context, i.e. the transition from the Disney Renaissance into the post-Renaissance period. In particular, passage #5 mentions The Emperor's New Groove (2000), which is directly situated after the time frame established in passage 1, thus forming a reasoning chain across the two. Similarly, passage #6 extends this timeline further, showcasing the evolution of Disney's productions.

---

**Query**

Q: Which film was released right after the Disney Renaissance period by Disney?

---

**Retrieved Passages**

**Round-1 Retrieval:**

1. In the history of The Walt Disney Company, the Disney Renaissance refers to the era from 1989 to 1999 during which Walt Disney Feature Animation (renamed Walt Disney Animation Studios in 2006) experienced a creative resurgence in producing successful animated films based on well-known stories, which restored public and critical interest in The Walt Disney Company as a whole.

2. Disney's The Little Mermaid is an American animated television series produced by Walt Disney Television Animation based on the 1989 Disney film of the same name.

3. Corso is a New York Foundation for the Arts Poetry Fellow and Sherwood Anderson Fiction Award Winner, and is the author of "Death by Renaissance" (2004).

---

**GPT**

The first paragraph is relevant and it's insufficient to answer the question. We need to know which Walt Disney films were released after 1999.

---

> **Retrieved Passages**
>
> **Round-2 Retrieval:**
>
> 4. The Apple Dumpling Gang is a 1975 American comedy-western film produced by Walt Disney Productions about a slick gambler named Russell Donovan (Bill Bixby) who is duped into taking care of a group of orphans who eventually strike gold during the California Gold Rush.
>
> 5. The Emperor's New Groove is a 2000 American animated buddy comedy film created by Walt Disney Feature Animation and released by Walt Disney Pictures. It is the 40th Disney animated feature film.
>
> 6. The Lion Guard is an American animated television series developed by Ford Riley based on Disney's 1994 film "The Lion King." The series was first broadcast with a television movie, titled The Lion Guard: Return of the Roar on Disney Channel on November 22, 2015, and began airing as a TV series on January 15, 2016, on Disney Junior and Disney Channel.

### A.2.2 EXPLICIT CONNECTION

Passages #4, #8, and #9 can be explicitly connected as they all refer to the same work, The Apple Dumpling Gang. Passage #4 introduces the original film, while Passage #8 mentions a drama produced by Walt Disney Productions, and Passage #9 references the film's composer. These connections are straightforward and fact-based, forming an explicit linkage through shared entities and production history. In our graph construction, such connections form strong edges that preserve concrete semantic relationships within the corpus.

> **Retrieved Passages through Keyword Extraction**
>
> **7.** *"Down in New Orleans"* is a jazz song from Disney's 2009 animated film *"The Princess and the Frog"*, written by Randy Newman. Several versions of the song were recorded for use in different parts of the film and other materials. The song was nominated for Best Original Song at the 82nd Academy Awards but lost to *"The Weary Kind"* from *"Crazy Heart"*.
>
> **8.** *Gun Shy* is an American sitcom that was shown on CBS from March 15 to April 19, 1983. The series, produced by Walt Disney Productions, was based on its popular comedy-western films: *"The Apple Dumpling Gang"* and *"The Apple Dumpling Gang Rides Again"*.
>
> **9.** Norman Dale "Buddy" Baker (January 4, 1918 – July 26, 2002) was an American composer who, together with Paul J. Smith, scored many Disney films, such as *"The Apple Dumpling Gang"* in 1975.

