# OpenReview forum: "HyperRAG: Query-Centric Retrieval Augmented Generation with Hyperbolic Structuring"
_ICLR.cc/2026/Conference — ICLR 2026 Conference Withdrawn Submission_

### Official Review · Reviewer_t4jh · 2025-10-30

**Soundness:** 2
**Presentation:** 3
**Contribution:** 2
**Rating:** 2
**Confidence:** 3

**Summary:**

This paper introduces HyperRAG, a query-centric retrieval system that builds upon traditional RAG architectures. The approach emphasizes mainly two points: Query-driven retrieval mechanisms to enhance the performance and accuracy of retrieval-augmented generation systems with dynamic updating.  Applying hyperbolic distance instead of Euclidean distance in the multi-hop QA RAG system.

**Strengths:**

S1: In my understanding, the paper not only utilizes traditional GraphRAG to enhance the model's reasoning capabilities, but also extracts implicit associations through text similarity recall at runtime. This query-time enhancement allows the knowledge graph to be updated over time.

S2: Compared to commonly used similarity metrics in text retrieval, such as Euclidean distance and cosine similarity, the paper introduces hyperbolic distance for multi-hop QA data. This distance metric can better distinguish similarity differences between text segments in long-hop chains, amplifying the significance of multi-hop differences. I appreciate the paper's thinking on this aspect.

**Weaknesses:**

W1: The core idea of the paper includes a streaming GraphRAG framework, which seems more like an engineering implementation that combines various tools together, lacking theoretical guidance. Following this approach, there could be many other streaming methods for updates - why must it be the approach mentioned in the paper?

W2: Although the paper provides a query-centric improvement solution, it may not be very practical in reality. Since the experimental dataset contains only 1,000 questions, this may present two problems in the context of real-world systems with large amounts of meaningless noise questions: 1. Will meaningless questions be identified as Implicit Connections, thereby degrading the quality of the dynamic knowledge graph? Are there any tests provided for this? 2. If there are a large number of query requests, does the method provide a trade-off between knowledge graph quality and cost? Are there related ablation experiments for this?

W3: The method is unclear (the pseudocode cannot even be considered an algorithm, but rather a process description). The experiments are insufficient and the experimental reproducibility details are not clear enough, lacking ablation experiments with sufficient methodological details, such as the impact of parameters on experiments. Unless I missed it, I did not see any description of reproducible materials in the paper. (Providing more details and ablation experiments could improve the rating)

**Questions:**

Q1: The hyperbolic distance mentioned in the paper is an interesting measure in mathematics. I understand this is not your contribution, but I indeed see this connection with multi-hop QA for the first time. However, your statement "While desirable in many metric spaces, these metrics universally satisfy the triangle inequality. As the number of hops increases (i.e., $p_1 \to p_2 \to \dots \to p_k$ ), passages with similar semantic embeddings have endpoints p0 and pk that remain close in the metric space. As k increases, the global semantic distance between distant nodes in the chain is underestimated." confuses me. How did you arrive at this conclusion? You have not rigorously stated such assumptions in your theory, nor have you proven the assumptions. I believe that in conventional representation models (including the BERT you mentioned), when building vector database indices and performing multi-hop vector retrieval, such assumptions do not necessarily exist (I have not seen relevant literature providing such insights).

Q2: Do you provide more experimental details or theoretical analysis elsewhere? Otherwise, as a complete submission, is the content too limited?

---

### Official Review · Reviewer_Aomi · 2025-10-30

**Soundness:** 2
**Presentation:** 2
**Contribution:** 2
**Rating:** 4
**Confidence:** 2

**Summary:**

The HyperRAG method presented in this paper is somewhat in the context of retrieval-augmented generation, and its effectiveness is demonstrated through experiments. However, there are clear areas for improvement.

**Strengths:**

The experimental design is well-thought-out, and the paper demonstrates HyperRAG's superior performance across multiple datasets, particularly in handling complex reasoning tasks. The motivation behind the method is clear, effectively integrating both explicit and implicit relationships.

**Weaknesses:**

**Clarity of the Paper:** In lines 185-192, the authors mention that the method is "driven by LLM-based understanding" and uses prompts to generate relationships. **While a toy prompt is shown in the Appendix, the real and comprehensive relationship extraction method remains unclear and needs to be described in more detail, which should show the robustness to chunk splitting noise in real scenarios.**

**Computational Costs in Relationship Extraction:** **The authors do not fully address the exponential growth of possible query-doc and doc-doc relationship combinations during relationship extraction, which incurs significant computational costs.** Although the authors performed partial analyses on tokens and processing time. **It is necessary to include a more comprehensive cost analysis and compare it with the baseline methods, especially under different corpus sizes. Without this comparison, discussing effectiveness alone is somewhat unfair, and the practical value of the method remains unclear.**

**Limitations of the Method:** The method relies on pre-labeled example queries. Without example queries, the system cannot operate effectively with just the corpus.

**Unsubstantiated Claims**:The discussion on hyperbolic embeddings in the paper is overly intuitive and lacks rigorous support. While the authors claim that larger hyperbolic distances lead to enhanced discriminative power and improved ranking stability, the provided distance statistics (e.g., Hyperbolic vs. Euclidean Max/Min/Mean for Top‑k) merely reflect the natural scale expansion of hyperbolic space, not an inherent advantage in distinguishing relevant from irrelevant entities. There is no evidence that inter-class contrast or retrieval ranking of borderline cases is actually improved. Moreover, the conclusions ignore the impact of curvature choice, normalization, and optimization stability. **To substantiate their claims, the authors should provide experiments analyzing inter/intra-class distance separation, ranking stability, and control for distance scale, rather than relying solely on the absolute magnitude of hyperbolic distances.**

**Questions:**

See Weaknesses.

---

### Official Review · Reviewer_gkgc · 2025-10-30

**Soundness:** 2
**Presentation:** 3
**Contribution:** 2
**Rating:** 4
**Confidence:** 4

**Summary:**

This paper proposes HyperRAG tailored to each query to address the shortcomings of previous query-agnostic and query-independent GraphRAG when latent relations are necessary since current knowledge graphs only capture explicit relations in priori. HyperRAG contains both explicit knowledge relationships as well as implicit connections guided by the given specific query. The experiments conducted have on relative datasets show a remarkable improvement compared with previous RAG baselines, showing the effectiveness of both explicit and implicit relations in RAG.

**Strengths:**

* This paper is well-written, the method is clear to understand.
* The method is simple and effective while keeping the cost equal or less compared with baselines.
* The experiments are thorough, including main results, ablation analysis, etc.

**Weaknesses:**

* In general, the implicit and underlying logistic relations in the proposed HyperRAG are somehow another formulation of CoT. Preliminary construction of the query-centric graph is like transforming textual CoT information into structured information stored in the graph. One of the baselines IRCoT can be formulated as RAG+textual CoT, and HyperRAG is like RAG+predefined structured CoT by offline relations construction by queries. Intuitively these two methods are similar but the advantage of HyperRAG against IRCoT is surprising. Could the authors give more explanations to this point?
* HyperRAG highly depends on the offline predefined implicit relations, when it comes to applications in real world, the advantages are not that decent. Instead, if CoT replaces the implicit relations given several explicit relations with query as centric node, the performance may achieve comparable performance.
* From Table 2, it is observed that the main improvements are from implicit relations, namely, reasoning results while stored as a graph. Previously getting the reasoning paths is like cheat, maybe the prompt engineering gives the answer directly.

**Questions:**

See in the weakness part.

---

### Official Review · Reviewer_MWB8 · 2025-10-31

**Soundness:** 3
**Presentation:** 2
**Contribution:** 3
**Rating:** 2
**Confidence:** 4

**Summary:**

This paper proposes HyperRAG, a query-centric retrieval-augmented generation framework that uses a dual-stage prompting strategy to build both explicit and implicit knowledge graphs, unifies them hierarchically, and learns their structure in hyperbolic space to enhance multi-hop reasoning and complex question answering.

**Strengths:**

1. This paper introduces a dual-stage prompting strategy that integrates explicit entity-level and implicit logical relations, enabling more query-centric graph construction.

2. This paper employs hyperbolic graph learning to effectively preserve hierarchical relationships and improve reasoning over multi-hop queries.

3. This paper demonstrates consistent performance gains across multiple benchmark datasets, showing both superior accuracy and computational efficiency.

**Weaknesses:**

1. The definition of the dual-stage prompting strategy is vague. The explicit entity-level connections are claimed to be derived from the LLM, but the details are not provided — in Section 3.1.1, only “lightweight keyword extraction techniques” are mentioned as the implementation approach, with no reference to LLM involvement. As a major contribution, a clear and explicit description is essential.

2. The implementation detail of tree fusion is not clear. The 3.1.3 section proposes a tree fusion mechanism that merges explicit and implicit knowledge trees, but no algorithmic formulations or pseudo-code are given.

3. The pruning action to prevent cycles in tree merging is questionable in design. This pruning operation may oversimplify the graph and reduce its representational capacity, potentially eliminating important semantic connections or multi-hop reasoning paths, thereby leading to the loss of critical knowledge that could have contributed to more comprehensive retrieval and reasoning.

4. The contribution of retrieval with learned hyperbolic embedding is not given. As mentioned in 3.1.3, "The final tree thus integrates both global knowledge priors and query-dependent reasoning chains", therefore, how about directly apply this tree-structural knowledge for answering?

5. Redundant design of extracted implicit connection. Semantic relationships are recoginized when "Building Query-guided Implicit Connection", like supporting, complement or exppanding, but they are not utlized in subsequent retireval in hyperbolic space.

6. The overall inference efficiency remains concerning. Because HyperRAG adopts a query-centric design, the graph must be dynamically built for each new query, making the graph construction cost non-negligible in real-time or large-scale applications. Ignoring this on-the-fly construction overhead during performance evaluation may overestimate the system’s practicality, as true end-to-end latency could be significantly higher.

**Questions:**

As discussed above in weakness.

---

### Note · Authors · 2025-11-21

I have read and agree with the venue's withdrawal policy on behalf of myself and my co-authors.